# Effects of the Addition of Flaxseed and Amaranth on the Physicochemical and Functional Properties of Instant-Extruded Products

**DOI:** 10.3390/foods8060183

**Published:** 2019-05-30

**Authors:** Jazmin L. Tobias-Espinoza, Carlos A. Amaya-Guerra, Armando Quintero-Ramos, Esther Pérez-Carrillo, María A. Núñez-González, Fernando Martínez-Bustos, Carmen O. Meléndez-Pizarro, Juan G. Báez-González, Juan A. Ortega-Gutiérrez

**Affiliations:** 1Facultad de Ciencias Químicas, Universidad Autónoma de Chihuahua, Nuevo Campus Universitario, Circuito Universitario, C.P. 31125 Chihuahua, Chih., Mexico; jazletobias@gmail.com (J.L.T.-E.); cmelende@uach.mx (C.O.M.-P); 2Departamento de Investigación y Posgrado, Facultad de Ciencias Biológicas, Universidad Autónoma de Nuevo León, Ciudad Universitaria, C.P. 66450 San Nicolás de los Garza, N.L., Mexico; numisamaya@hotmail.com (C.A.A.-G.); maria.nunezgn@uanl.edu.mx (M.A.N.-G.); juan.baezgn@uanl.edu.mx (J.G.B.-G.); 3Centro de Biotecnología-FEMSA, Escuela de Ingeniería y Ciencias, Tecnológico de Monterrey, Av. Eugenio Garza Sada 2501 Sur, C.P. 64849 Monterrey, N.L., Mexico; perez.carrillo@tec.mx; 4Centro de Investigación y de Estudios Avanzados del Instituto Politécnico Nacional Unidad-Querétaro, Libramiento Norponiente 2000, Fracc. Real de Juriquilla, C.P. 76230 Santiago de Querétaro, Qro., Mexico; fmartinez@cinvestav.mx; 5Facultad de Zootecnia y Ecología, Universidad Autónoma de Chihuahua, Periférico R. Almada Km 1, C.P. 33820 Chihuahua, Chih., Mexico; jortega@uach.mx

**Keywords:** extruded products, flaxseed, amaranth, dietary fiber, extrusion-cooking

## Abstract

The addition of flaxseed and amaranth on the physicochemical, functional, and microstructural changes of instant-extruded products was evaluated. Six mixtures with different proportions of amaranth (18.7–33.1%), flaxseed (6.6–9.3%), maize grits (55.6–67.3%) and minor ingredients (4.7%) were extruded in a twin-screw extruder. Insoluble and soluble fiber contents in extrudates increased as the proportions of amaranth and flaxseed increased. However, the highest flaxseed proportion had the highest soluble fiber content (1.9%). Extruded products with the highest proportion of flaxseed and amaranth resulted in the highest dietary fiber content and hardness values (5.2 N), which was correlated with the microstructural analysis where the crystallinity increased, resulting in larger, and more compact laminar structure. The extruded products with the highest maize grits proportion had the highest viscosity, expansion, and water absorption indexes, and the lowest water solubility index values. The mixtures with amaranth (18.7–22.9%), flaxseed (8.6–9.3%), and maize grits (63.8–67.3%) resulted in extruded products with acceptable physicochemical and functional properties.

## 1. Introduction

Currently, an increasing trend in the demand for processed foods that include pro-health compounds such as soluble fiber is occurring due to evidence of potential health benefits to consumers. Reduction in various types of chronic diseases such as cancer, cardiovascular disease, type II diabetes, and various gastrointestinal disorders are among them [1]. The development of products and processes that incorporate high-fiber ingredients without altering the physical, functional, and sensory properties of the processed foods are of interest and desirable to meet current consumer trends. A technological alternative for the incorporation of ingredients in processed products is the extrusion-cooking process, a very versatile technique widely used for the development of breakfast cereals and instant foods [2]. Some extruded products marketed as breakfast cereals have significant caloric value. One of the strategies that has allowed the food industry to reduce the energetic density and produce more healthy products has been the incorporation of dietary fiber. However, the addition of dietary fiber during extrusion, especially if it is insoluble, results in products with less expansion and crispness and a higher bulk density and hardness, which are properties less preferred by consumers [3]. These characteristics can be explained by the interactions of fiber with starch that impact the mechanisms of starch gelatinization. These, in turn, are related to water absorption of the extrudates and other physicochemical transformations that occur during extrusion, such as viscoelastic properties associated with the stabilizing membranes of the bubbles formed during bubble growth in the final product [4]. Adding insoluble fiber to extruded products has been shown to decrease the proportion of starch in the food matrix, thereby reducing the water absorption capacity and, in turn, the viscosity caused by the gelatinization of the starch, which results in a reduction in the expansion of the final product [3]. A sectional reduction in the expansion of extruded products had been reported by Brennan et al. [5], due to an increase in insoluble fiber results in structures with a high number of small cells and a high cell density. Several authors reported that the bulk density is increased by adding insoluble fiber to extruded products [5,6]. Also, it has been reported that extrusion process causes significant effects on the dietary fiber content, as breakage of structural polysaccharides or complex carbohydrates formation [3]. These effects include the formation of resistant starch that may occur during extrusion and the formation of covalent interactions between macronutrients and insoluble components (such as insoluble fiber) that make the extrudates indigestible by amylase or protease activity [7]. 

The development of ready-to-eat products with high fiber content (6 g of fiber/100 g of product) [8], is based on the use of ingredients that meet this requirement and at the same time, provide numerous health benefits. Flaxseed and amaranth have been used specifically in extruded products due to their health benefits for consumers, but, usually, they are used individually [9,10,11,12,13]. Amaranth contains a good balance of amino acids, including the essential amino acid lysine, which is present in limited amounts in most cereals [14,15]. Flaxseed is low in carbohydrates (sugars and starches), high in fiber and protein, and rich in polyunsaturated fatty acids, particularly alpha-linolenic acid (ALA or ALN) and linoleic acid (AL), known as omega-3 and omega-6 essential fatty acids, respectively [16,17]. An important component of these two grains is soluble fiber, considered a functional ingredient because it generates high-viscosity products by causing the gelation of chyme. Chyme acts as a network to capture glucose and cholesterol molecules in their passage through the gut, hindering their absorption and thereby decreasing blood glucose and cholesterol levels, resulting in beneficial health effects [18].

Despite the health benefits of amaranth and flaxseed, little information has been reported on the combined effect of both ingredients on the physicochemical and functional properties of extruded products. Therefore, the aim of this study was to develop an extruded product with high-fiber ingredients and to evaluate the effects of the addition of flaxseed and amaranth on the physicochemical and functional properties of instant-extruded products. 

## 2. Materials and Methods 

### 2.1. Materials 

Grains of amaranth (*Amaranthus hypochondriacus* L.), flaxseed (*Linum usitatissimum* L.) and minor ingredients such as sucralose, cocoa, and cinnamon were obtained from a local distribution store (Chihuahua, Chihuahua, Mexico). Also, maize grits number 4 (GPC, Muscatine, IA, USA) was used for the extrusion food matrix. The amaranth and flaxseed grains were milled in a roller mill (Zhengzhou Chengli Grain & Oil Machinery Co., Ltd., model 6F-2240, Zhengzhou, China), separately and sieved in a mesh number 35 (model RX-24, Tyler industrial products, Mentor, OH, USA) to obtain a particle size of 0.5 mm. All materials were stored in plastic bags at room temperature until their use.

### 2.2. Chemicals

Hydrochloric acid 37.2%, sulfuric acid 97.9%, hexane 99.8%, ethanol 99.9%, and boric acid 99.5% were all analytical grade and obtained from J.T. Baker (Mexico City, Mexico). Analytical grade sodium hydroxide (97.0%) was obtained from Sigma-Aldrich (St. Louis, MO, USA). The selenium reaction mixture was obtained from Merck (Darmstadt, Germany). The kit for soluble and insoluble dietary fiber was obtained from Sigma-Aldrich (St. Louis, MO, USA) [19]. 

### 2.3. Methods

#### 2.3.1. Mixtures Preparation

Different proportions of amaranth and flaxseed flours were mixed with maize grits and fixed minor ingredients (sucralose, cocoa, and cinnamon). The ingredients were mixed in an industrial mixer (Bathammex, Mexico City, Mexico) for five minutes obtaining six different mixtures as shown in Table 1. The proportion of each ingredient in each mixture was determined considering fat and crude fiber contents around 5.6% and 2.5%, respectively; this according to the literature [2,20] to obtain extruded products with acceptable physicochemical and sensory characteristics. 

#### 2.3.2. Extrusion Process

For the extrusion process, we used a twin-screw corotating extruder (BCTM-30, Bühler, AG, Uzwil, Switzerland) with a 600 mm length, a length to diameter ratio (L/D) of 20:1, a die opening of 4 mm, the screw configuration was selected specifically to create high levels of shear. The mixtures were fed to the extruder at a rate of 7.5 kg/h and were processed at a speed of 272 rpm at moisture content of 0.22 kg water/kg dry matter, which was adjusted within the extruder, at a temperature of 150 °C. It was controlled at the final stage of the extruding chamber by using a TT-137N water heater (Tool-temp, Sulgen, Switzerland). All extrudates were dried at 120 °C for 15 min in an air convection oven (Electrolux 10 GN/1, Stockholm, Sweden) at air cross-flow velocity of 1.5 m·s^-1^ until the extrudates reached a range moisture level of 0.017–0.031 kg H_2_O·kg ss^−1^. The extrudates were packed and stored at room temperature (25 °C) until evaluation.

### 2.4. Analytical Methods

#### 2.4.1. Proximate Analyses

The starting materials and extruded products were analyzed for moisture, protein, fat, crude fiber, and ash content according to methods 950.02, 960.52, 920.39, 962.09, and 923.03 of AOAC [19], respectively. Carbohydrates mass was calculated by difference. The analyses were carried out in triplicate for each treatment, and the results were expressed in g/100 g.

#### 2.4.2. Insoluble and Soluble Dietary Fiber

The insoluble and soluble fiber in the ingredients, the mixtures and the extruded products were determined with the total dietary fiber assay kit (Sigma-Aldrich, St. Louis, MO, USA) according to method 991.43 of AOAC [19]. The analyses were carried out in triplicate for each treatment, and the results were expressed in g/100 g.

### 2.5. Functional Properties of the Extruded Products

#### 2.5.1. Water Absorption and Water Solubility Indexes

The water absorption index (WAI) and water solubility index (WSI) were determined in triplicate following the procedures described by Anderson et al. [21]. The methods measure the quantity of water incorporated in the flour and the soluble solids that dissolve in water at 30 °C. Samples were weighed (2.5 g) into plastic tubes and mixed with 30 mL of distilled water. The samples were manually shaken, the slurries were centrifuged for 10 min at 3200× *g* (Thermo IEC model CL3-R, Thermo Scientific, Waltham, MA, USA), and the supernatant was decanted into pre-weighed porcelain capsules. Capsules were dried for 24 h at 105 °C and weighed. The gel remaining in the tubes after decanting the supernatant was weighed. The ratio between gel-forming solids and soluble solids was measured as grams of water per gram of flour. The WAI was calculated as a percentage of remaining gel weight compared to the pre-dried weight from the extruded products. The WSI was calculated as a percentage of the dried supernatant weight compared to the pre-dried weight from the extruded products.

#### 2.5.2. Bulk Density

The bulk density (BD) was determined according to Jin et al. [22] in which the ground extrudate (40/60 mesh) was poured into a cylindrical container. Excess extrudate was scraped off, and the net weight of the powder was divided by the volume of the container. Bulk density was expressed in kilograms per liter (kg L^−1^). The analysis was performed in triplicate, and mean values were reported.

#### 2.5.3. Expansion Index

The expansion index (EI) was reported as the ratio of extruded product diameter and the diameter of the die hole [23]. Values were reported as means of 60 measurements.

### 2.6. Physical Properties of the Extruded Products

#### 2.6.1. Textural Measurement: Hardness and Crispness

The evaluation of the hardness and crispness of the extrudates was performed according to the method described by Ding et al. [24], and carried out using a Texture Analyzer TA.XT (Texture Technologies Corporation, Scarsdale, New York/Stable Micro Systems, Haslemere, Surrey, UK) configured with a 2 mm punch at a crosshead speed of 5 mm/s and a travel distance of 15 mm. Twenty-four extruded unit samples were taken randomly from each treatment and analyzed. A force time curve was recorded and analyzed by the Texture Exponent 32 (Surrey, UK) program to calculate the maximum force (N) to determine the hardness and the area under the curve (N/mm) to determine the crispness. 

#### 2.6.2. Pasting Properties of the Extruded Products

The amylographic viscosity profile was determined according to Sánchez-Madrigal et al. [25], with some modifications, using a Rapid Visco Analyzer (RVA SUPER 4 (Newport Scientific, Sydney, Australia). Flour sample suspensions were prepared by weighing 4 g of milled and dried (50 ± 2 °C, 12 h) extrudates with a 7.5 to 8.5% moisture content and a small particle size (0.25 mm) into an RVA canister and individually adjusting each sample to a total weight of 28 g using distilled water. The rotating paddles were held at 50 °C for 1 min to stabilize the temperature and ensure uniform dispersion and heated to 92 °C at a rate of 5.6 °C/min, which was held constant for 5 min. The dispersion was cooled to 50 °C at the same rate and was held at 50 °C for 1 min. The maximum viscosity (MaxV) at 92 °C, the minimum viscosity (MinV or lowest viscosity at the end of heating constant period at 92 °C) and the final viscosity (FinV attained during cooling to or holding at 50 °C) were recorded. The total setback viscosity or viscosity of retrogradation (final viscosity minus minimum viscosity) was calculated from these parameter values. The viscosity with RVA was obtained in RVU units (1 RVU = 10 centipoises). Each treatment was performed twice. 

### 2.7. Scanning Electron Microscopy

This analysis was performed according to the method described by Sánchez-Madrigal et al. [25]. Flours of each extruded cereal with a particle size <0.15 mm and a moisture content of 1% were stuck to stubs and coated with a gold layer in a high vacuum using a Denton vacuum evaporator (Desk II), set to a pressure of 7.031 × 10^−2^ kg cm^−2^. The samples were examined using a scanning electron microscope (JSM-5800LV, JEOL, Akishima, Japan) equipped with a secondary electron detector at an acceleration rate of 10 kV. 

### 2.8. Statistical Methods

A univariate analysis of variance was performed adjusting a model that included the main effects and their interaction (Minitab 16). When the effect of the interaction factor or the main effects was significant (α 0.05); means comparison was performed by Tukey’s test [26].

## 3. Results and Discussion

### 3.1. Raw Materials Characterization

Proximate analysis showed a significant difference (*p* < 0.05) between the raw materials for each of the components (Table 2) and indicated that they were of high nutritional value. Is important to highlight that flaxseed had the highest protein, fat, and fiber content, with the lowest carbohydrate content. Whereas the amaranth had protein content twice as maize grits. These values (%) are consistent with those reported in the literature [16,27], where amaranth contains a good balance of amino acids, including lysine, an essential amino acid that is not found in most cereals [14]. Flaxseed has been reported to be low in carbohydrates (sugars and starches), high in quality protein, fiber and rich in polyunsaturated fatty acids [16]. Maize grits were the main source of carbohydrates, as shown in Table 2. Meanwhile amaranth and flaxseed showed the highest dietary fiber contents (Table 3), these results agree with Morris [16] and Cervantes [27]. The soluble fiber content in the ingredients was in the following relevance order: flaxseed (9%), amaranth (1.3%) and maize grits (0.71%).

### 3.2. Extrudate Characterization

Proximate analysis of different extruded products is shown in Table 2. Chemical characteristics were significantly affected (*p* < 0.05) for amaranth and flaxseed additions. The extruded products had high percentage of protein compared to other commercial extruded cereals, which typically have protein content between 5 and 8% [28]. This is due to the contributions of protein of amaranth (17.4%) and flaxseed (22.4%) (Table 2). Similar protein content for extruded amaranth with maize grits were reported [10]. The addition of amaranth could lead to a good balance of amino acids because it contains lysine, an essential amino acid, which is not found in most cereals [14]. Whereas flaxseed protein is rich in arginine, aspartic acid and glutamic acid and deficient in lysine [16].

The extrudates had a desirable crude fiber content (<2%; Table 2), with acceptable physicochemical and sensory characteristics for the consumer without a negative effect on the caloric and nutrient content, which could especially benefit young children [2]. Additionally, the fat content in all extruded products was below the minimum acceptable value of 5.6% [20] to reach desirable characteristic in extruded products. From the nutritional point of view (<5%) a low-fat product was obtained.

### 3.3. Dietary, Insoluble and Soluble Fiber 

Table 3 shows the soluble and insoluble fiber content of the different mixtures before being subjected to the extrusion process. The addition of flaxseed significantly affected the soluble and total dietary fiber content. After the extrusion process of the mixtures, the results for the dietary fiber content showed a significant difference between treatments (*p* < 0.05), indicating that variation in the proportions of ingredients (amaranth and flaxseed) and their interaction, significantly affected the content of insoluble and soluble fiber in extruded products (Figure 1). Additionally, it was observed that the extrusion process caused an increase in the soluble fiber content and a decrease in insoluble fiber compared to the non-extruded mixtures (Table 3, Figure 1). The extruded products from the mixtures 3, 4, and 5 had the highest percentage of soluble fiber but they did not present a significant difference. This can be explained by the fact that extrudates 4 and 5 had the highest content of flaxseed (9.3% and 8.6% respectively), which is a significant source of soluble fiber, reaching up to 9% (Table 3). In turn, extrudate 3 had the highest percentage of amaranth and flaxseed (41.7%), which resulted in high soluble fiber content. The rest of the extrudates (mixtures 1, 2 and 6) had a low percentage of soluble fiber without showing significant differences among them (Figure 1) because they had low percentages of flaxseed. Various studies had shown that dietary fiber, especially soluble fiber in extrudates, increases when they are subjected to the extrusion process [29]. Additionally, other biomolecules such as starch undergo structure changes, leading to the formation of resistant starch, another possible mechanism causing the increase in fiber during the extrusion process [7]. On the other hand, total dietary fiber content could decrease due to the fact that during the extrusion process, shear stress caused by high screw speed, combined with high process temperatures causes chemical bond breakage from complex carbohydrates, releasing molecules as xylose, glucose, arabinose, oligosaccharides, and, preferentially, slightly branched arabinoxylans which are solubilized [29,30]. Similarly, this decrease in total dietary fiber content could be observed in some of our treatments.

A food product is considered high in dietary fiber when it contains >6% [8]. Therefore, the extruded products presented high fiber content (Figure 1), due to the addition of amaranth and flaxseed, which contain a high percentage of dietary fiber; 13.1 and 59.6% respectively (Table 3).

### 3.4. Water Absorption and Water Solubility Indexes

The WAI in the extruded products was significantly affected (*p* < 0.05) by the addition of flaxseed and amaranth and their interaction, in the mixtures (Table 4). The extruded cereal from mixture 4 (9.3% flaxseed, 18.7% amaranth, and 67.3% maize grits) resulting in high WAI values due to its high maize grits (cornstarch) content which, during the extrusion process, undergoes pronounced changes in gelatinization properties, favoring a higher water absorption. This is consistent with the amylographic viscosity profile, where the extrudates with the highest values of viscosity have the highest values of WAI (Figure 2). On the other hand, low WAI values resulted for extrudate from mixture 3 (6.6% flaxseed, 33.1% amaranth, and 55.6% maize grits), which had the lowest maize grits (cornstarch) content but the highest proportion of high-fiber ingredients (39.7% amaranth and flaxseed). Similar results were obtained in extrudates from mixtures 2 and 6 (Table 4). Similar effects of fiber addition on extruded products were reported by Altan et al. [31] for the extrusion of barley mixtures with tomato pomace. 

WSI is an indicator of the degradation of molecular components: an example is the amount of soluble polysaccharide of starch released after extrusion, which is a measurement of the degree of conversion of starch during extrusion [24]. Table 4 shows significant effects on the WSI of the extruded products by the addition of amaranth and flaxseed, in the mixtures. Extrudates from mixtures 2, 5, and 6, with a high fiber content (Table 4), had the highest WSI values (0.5). Whereas the extrudate from mixture 1, with a low fiber content, and the extrudate from mixture 4, with a high starch content, had the lowest WSI values (0.46 and 0.45, respectively). The increase in the fiber content caused an increase in the WSI values, which can be attributed to the rupture of the structural polysaccharides by the extrusion process [29,30]. A similar finding was reported by Ganorkar et al. [13] and Altan et al. [31].

### 3.5. Bulk Density

The bulk density in extruded cereals shows significant changes (*p* < 0.05) due to the addition of the ingredients in the mixtures (Table 4). The extrudate from mixture 6, with a high fat content, presented the lowest density value: this can be attributed to the fat’s low-density and the oil contained in cereals, which are emulsified during extrusion due to the high pressure reached during the process. The fine drops of fat are coated by starches and proteins, leaving the fat encapsulated and causing a decrease in the density [32]. The extruded products from the mixtures 2 and 4 were high in protein and presented the highest density values. The rigid tertiary structure, high cohesiveness, high molecular weights, and structural functions in cereal proteins such as corn, can increase the density of food products [33]. Ryu et al. [23], reported that the density of an extruded product is strongly affected by water, fiber, fat, and starch content. The extruded products from the mixtures 6 and 3, with similar amaranth contents (26.4 and 33.1 g/100 g respectively), resulted in the lowest density values (4.6 and 4.7 kg L^−1^, respectively; Table 4). These results were consistent with those found by Ilo et al. [9], who evaluated the effect of extrusion-cooking process on the properties of extruded rice flour and amaranth blends. They observed that amaranth had an important influence on the product density, resulting in a minimum density value at amaranth content of 30 g/100 g. Another report had shown increases in the bulk density values during the extrusion of rice flour and corn fortified with flaxseed [12].

### 3.6. Expansion Index

The addition of the different ingredients in their different proportions, and their interaction, significantly affected the expansion of extruded products (*p* < 0.05) (Table 4). The EI of instant-extruded products is very important since it is directly related to consumer acceptability; related typically to an inflated, lightweight and crunchy structure [34], mainly attributed to the presence of starch in the final extruded products [35]. The extrudate of the mixture 4 resulted in the highest (*p* < 0.05) EI (3.33), followed by the extruded mixture 5 (3.17) due to a higher percentage of maize grits (starch), and finally the extruded products containing less maize grits (mixtures 1, 2 and 6) did not have significant difference between them and they presented the lowest EI values (Table 4). It is important to note that the mixtures 4 and 5 (with higher expansion values), contained a higher starch amount and a lower percentage of the ingredients with high dietary fiber content: amaranth and flaxseed. Mixtures 1, 2, and 6 with higher amaranth and flaxseed content presented the lowest EI, due to their high dietary fiber content, which affects the expansion of extruded products. The effect of the fiber on the expansion of the extruded products depends mainly on its interactions with the starch and, therefore, on the type and amount of fiber. Insoluble fiber significantly reduces expansion volumes and increases the density of extruded products. Conversely, soluble fiber leads to better expansion volumes, unaffecting the bulk density of the extruded products compared to the insoluble fiber components [3]. The difference in expansion behavior between soluble and insoluble fiber can be explained by their interactions with starch, differences in water absorption and plasticizing behavior, but also by the physicochemical transformations they undergo during extrusion [3]. This is consistent with the results reported by Altan et al. [31], who made extruded barley using tomato pomace as fiber source, noting that additions of tomato pomace, provoked a decrease of the EI on the final products. Similar results were reported by other authors [35,36,37].

### 3.7. Textural Measurement (Hardness and Crispness)

An important quality parameter of ready-to-eat extrudates is texture. Table 4 shows the values of hardness and crispness of extruded products made from the different mixtures of ingredients. The crispness of the extruded products was not significantly different (*p* > 0.05) among the various mixtures. Similar findings have been reported for corn extruded with amaranth, where the amaranth content from 20 to 35% had no substantial effect on crispiness [11]; this amaranth percentage was close to the one used in this study (18.7–33.1%).

However, the hardness of the cereals was significantly affected (*p* < 0.05) by the added ingredients (flaxseed and amaranth) and their interaction. Extruded products from mixture 1and 5 had the highest hardness (*p* < 0.05) attributed to their high dietary fiber content. As was described above, the addition of dietary fiber leads to reduced expansion volumes and increases in density of the extruded products, inducing harder textures and less crispiness [3]. This result is consistent with the results reported by Ganorkar and Jain [12], who showed that an increase in added flaxseed caused an increase in the hardness of the extruded products. A similar finding was reported by Brennan et al. [5] where they showed that increases in the wheat bran content up to 15% in extruded breakfast cereals causes breaking force increases. In contrast, it has also been reported that soluble fibers, such as inulin, deliver a more favorable texture compared to insoluble fibers, such as bran fiber [6]. This was corroborated by Brennan et al. [5], who observed a slight hardness change when adding either inulin or guar gum to extruded corn flour. This can be corroborated by our results, where the extruded products from mixture 3 and 4 (Figure 1) contained the highest percentage of soluble fiber and the lowest hardness values (4.7 N; Table 4). 

### 3.8. Pasting Properties of the Extruded Products

The addition of amaranth and flaxseed to the mixtures significantly affected (*p* < 0.05) the amylographic viscosity profile (RVA) of the extruded products (Table 4). MaxV is the peak viscosity where the highest degree of starch gelatinization occurs. The MaxV values obtained for each of the extruded products were very low, due the damage in the starch granules during the extrusion process [38]; this led to a notorious decrease of viscosity values in the extruded products, as shown in Figure 2. A similar trend on other viscosity parameters (MinV, FinV, and setback viscosity) values is shown in Table 4. The mixture 4, with the highest starch content (67.3% maize grits), had the highest MaxV value (*p* < 0.05); whereas mixture 3, with the lowest percentage of starch (55.6% maize grits), had the lowest value (*p* < 0.05). On the other hand, it is possible that the formation of complex structures during extrusion-cooking through interactions between starch-lipid complexes and/or starch-protein walls prevents adequate gelatinization of the starch [39,40]. Another factor influencing the pasting properties of extruded products is the presence of dietary fiber, which leads to a decrease in the fraction of water-swelling starch, due to its replacement by the fiber [37]. All these factors limited a complete starch gelatinization, causing a decrease in viscosity. Similar results were observed in a study where increases in amaranth in rice-amaranth blends generally decreases the viscosity of extruded products [9]. Similar findings were reported for extruded cornstarch blends with whey protein concentrate and Agave tequilana fiber [37].

Additionally, low setback values were found for all extruded products, indicating low rates of starch retrogradation and syneresis. During cooling, reassociation of starch molecules, especially amylose, result in viscosity increase favoring the final viscosity. This phase is commonly described as the setback region during which retrogradation and reordering of starch molecules occurs [41].

### 3.9. Scanning Electron Microscopy

The scanning electron micrographs revealed the impact of the different ingredients (Figure 3). After extrusion, it can be observed that combination of shear force and temperature inside the barrel caused microstructural changes in the extrudates of the different treatments [42]. The microstructural analysis shows that the addition of amaranth and flaxseed, increased the fiber content in the mixtures, resulting in compact agglomerates, increased crystallinity, and larger, more compact laminar structures (Figure 3c–f). This can be attributed to high protein (12%) and fiber (9–13%) content in the extrudates. Fiber tends to rupture cell walls and promotes breakage of air cells during extrusion, which prevents matrices from expanding [43], resulting in harder textures, higher densities and more compact structures as shown in the micrographs (Figure 3c–f). Similar results were found by Zhang et al. [29] and Cueto et al. [42].

## 4. Conclusions

This research shows that different levels of amaranth and flaxseed in the development of extruded products had a significant impact on their functional and physicochemical properties. The extruded products obtained had high protein content (>12%), which is higher than in the commercial breakfast cereals. Besides these characteristics, the obtained extruded products presented a healthy fat content (<5%) and a high content of soluble and insoluble dietary fiber. 

Another important ingredient was the maize grits, which was the source of starch, and it served as a basis to produce some expansion level in the extrudates. Extruded products with low levels of starch (maize grits) and high levels of fiber in the mixtures resulted in extrudates with low EI and high hardness values. These results suggest that the extrusion-cooking process of high-fiber flours such as flaxseed (8.6–9.3%) and amaranth (18.7–22.9%) in mixture with maize grits (63.8–67.3%) and minor ingredients results in an extruded product of good nutritional quality, with suitable functional and physicochemical characteristics.

## Figures and Tables

**Figure 1 foods-08-00183-f001:**
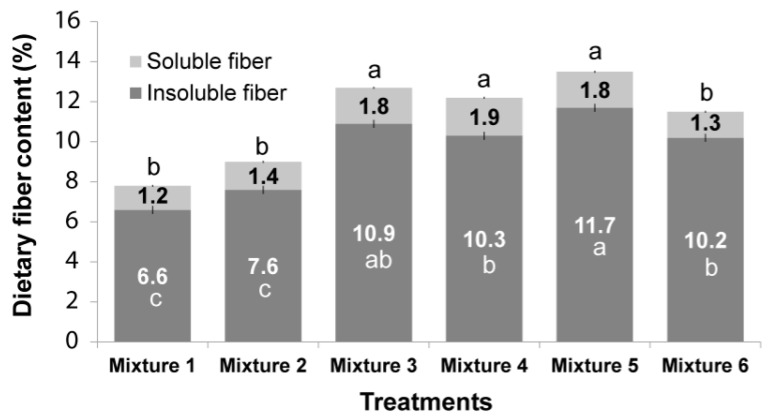
Insoluble and soluble dietary fiber content of extruded products. Means ± standard error (SE). SE insoluble fiber, 0.2; SE soluble fiber, 0.05. Means by columns and colors with different letters show significant differences based on contrast tests (*p* < 0.05).

**Figure 2 foods-08-00183-f002:**
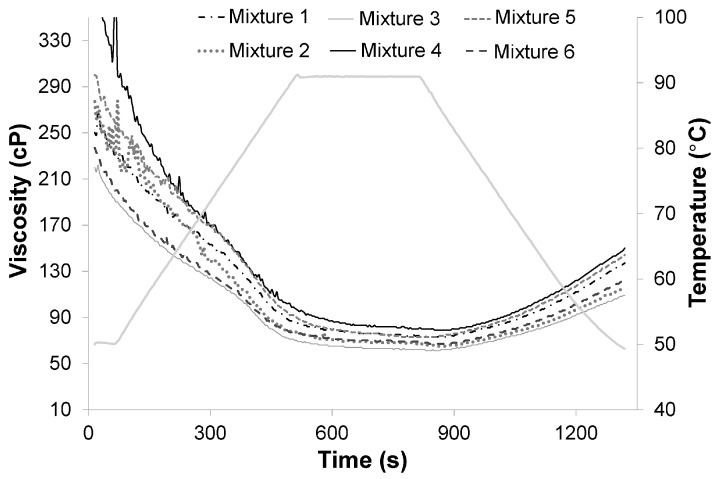
Amylographic viscosity profile of extruded products obtained with Rapid Visco Analyzer (RVA).

**Figure 3 foods-08-00183-f003:**
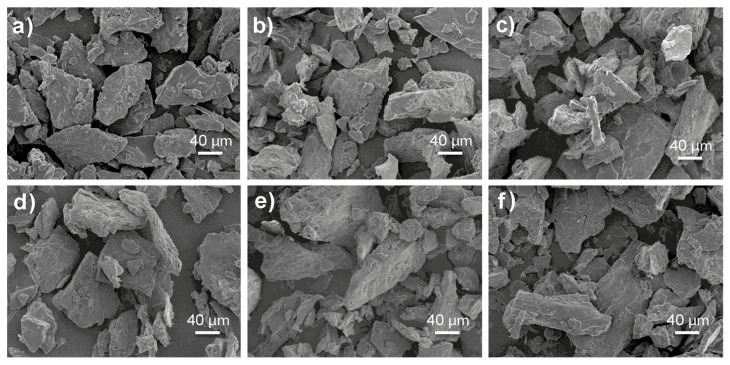
Micrographs of the extruded products from: (**a**) Mixture 1, (**b**) Mixture 2, (**c**) Mixture 3, (**d**) Mixture 4, (**e**) Mixture 5, (**f**) Mixture 6.

**Table 1 foods-08-00183-t001:** Proportion of ingredients of the different mixtures *.

Treatment	Flaxseed (%)	Amaranth (%)	Maize grits (%)	Minor Ingredients (%)
Mixture 1	8.2	24.7	62.4	4.7
Mixture 2	7.3	29.3	58.7	4.7
Mixture 3	6.6	33.1	55.6	4.7
Mixture 4	9.3	18.7	67.3	4.7
Mixture 5	8.6	22.9	63.8	4.7
Mixture 6	7.9	26.4	61	4.7

* Minor ingredients: sucralose (1.6%), cocoa (2.5%), cinnamon (0.6%).

**Table 2 foods-08-00183-t002:** Proximate composition of the raw materials and the extruded products *.

Component (%)	Amaranth	Flaxseed	Maize grits	Mixture 1	Mixture 2	Mixture 3	Mixture 4	Mixture 5	Mixture 6
Moisture	1.4 ± 0.01 ^c^	5.3 ± 0.01 ^b^	11.3 ± 0.01 ^a^	3.0 ± 0.13 ^a^	2.1 ± 0.13 ^bc^	2.4 ± 0.13 ^ab^	3.1 ± 0.13 ^a^	1.7± 0.13 ^c^	2.8 ± 0.13 ^ab^
Crude Fat	7.1 ± 0.18 ^b^	37.2 ± 0.18 ^a^	0.99 ± 0.18 ^c^	2.6 ± 0.06 ^b^	2.7 ± 0.06 ^b^	2.6 ± 0.06 ^b^	2.7 ± 0.06 ^b^	2.4 ± 0.06 ^b^	3.0 ± 0.06 ^a^
Crude Fiber	3.2 ± 0.05 ^b^	16.3 ± 0.05 ^a^	0.64 ± 0.05 ^c^	1.5 ± 0.05 ^b^	1.7 ± 0.05 ^ab^	1.9 ± 0.05 ^a^	1.7 ± 0.05 ^ab^	1.7 ± 0.05 ^ab^	1.9 ± 0.05 ^a^
Ash	3.0 ± 0.03 ^b^	3.3 ± 0.03 ^a^	0.55 ± 0.03 ^c^	1.6 ± 0.02 ^c^	1.7 ± 0.02 ^ab^	1.8 ± 0.02 ^a^	1.6 ± 0.02 ^c^	1.6 ± 0.02 ^c^	1.7 ± 0.02 ^b^
Crude Protein	17.4 ± 0.2 ^b^	22.4 ± 0.2 ^a^	8.7 ± 0.2 ^c^	12.2 ± 0.08 ^ab^	12.4 ± 0.08 ^a^	12.0 ± 0.08 ^bc^	12.3 ± 0.08 ^ab^	11.7 ± 0.08 ^c^	12.3 ± 0.08 ^ab^
Carbohydrates	67.9	15.5	77.8	79.1	79.4	79.3	78.6	80.9	78.3

* Means ± standard error (SE). Means by files for raw materials and extruded products, with different letters show significant difference, contrast test (*p* < 0.05). Carbohydrates were calculated by difference.

**Table 3 foods-08-00183-t003:** Dietary fiber content of the raw materials and the mixtures without extruding *.

Component (%)	Amaranth	Flaxseed	Maize grits	Mixture 1	Mixture 2	Mixture 3	Mixture 4	Mixture 5	Mixture 6
SDF	1.3 ± 0.23 ^b^	9.0 ± 0.23 ^a^	0.71 ± 0.23 ^b^	0.6 ± 0.1 ^b^	1.3 ± 0.1 ^a^	1.3 ± 0.1 ^a^	1.5 ± 0.1 ^a^	1.6 ± 0.1 ^a^	1.2 ± 0.1 ^a^
IDF	11.9 ± 0.15 ^b^	50.6 ± 0.15 ^a^	7.3 ± 0.15 ^c^	9.3 ± 0.05 ^cd^	9.7 ± 0.05 ^c^	8.9 ± 0.05 ^d^	11.6 ± 0.05 ^b^	12.0 ± 0.05 ^b^	12.6 ± 0.05 ^a^
TDF	13.2 ± 0.32 ^b^	59.6 ± 0.32 ^a^	8.0 ± 0.32 ^c^	9.9 ± 0.11 ^d^	11.0 ± 0.11 ^c^	10.2 ± 0.11 ^d^	13.1 ± 0.11 ^b^	13.6 ± 0.11 ^ab^	13.8 ± 0.11 ^a^

* Means ± standard error (SE). Means by files for raw materials and mixtures without extruding, with different letters show significant difference, contrast test (*p* < 0.05). SDF, soluble dietary fiber; IDF, insoluble dietary fiber; TDF, total dietary fiber.

**Table 4 foods-08-00183-t004:** Functional and physical properties of the extruded products *.

Treatments	BD (kg L^−1^)	EI	WSI	WAI	Hardness (N)	Crispness (N/mm)	MaxV (cp)	MinV (cp)	FinV (cp)	Setback Viscosity (cp)
Mixture 1	0.5 ^a^	3.11 ^c^	0.46 ^bc^	2.9 ^b^	5.0 ^ab^	26.9 ^a^	86.2 ^b^	69.9 ^abc^	151.0 ^a^	81.1 ^a^
Mixture 2	0.49 ^abc^	3.10 ^c^	0.5 ^a^	2.5 ^c^	4.9 ^bc^	25.4 ^a^	75.3 ^bc^	65.7 ^bc^	124.5 ^bc^	58.9 ^c^
Mixture 3	0.47 ^cd^	3.06 ^d^	0.49 ^ab^	2.5 ^c^	4.7 ^c^	24.9 ^a^	66.7 ^c^	63.2 ^c^	111.1 ^c^	48.0 ^c^
Mixture 4	0.49 ^ab^	3.33 ^a^	0.45 ^c^	3.4 ^a^	4.7 ^c^	25.7 ^a^	99.3 ^a^	79.8 ^a^	162.5 ^a^	82.7 ^a^
Mixture 5	0.47 ^bcd^	3.17 ^b^	0.5 ^a^	2.9 ^b^	5.2 ^a^	26.7 ^a^	85.6 ^b^	75.2 ^ab^	154.8 ^a^	75.1 ^ab^
Mixture 6	0.46 ^d^	3.09 ^c^	0.5 ^a^	2.5 ^c^	4.8 ^bc^	26.1 ^a^	76.1 ^bc^	70.2 ^abc^	130.5 ^b^	60.5 ^bc^

* Means ± standard error (SE). SE expansion index, 0.007; SE bulk density, 0.006; SE WSI, 0.007; SE WAI, 0.05; SE Hardness, 0.16; SE Crispness, 1.3; SE MaxV, 2.4; SE MinV, 2.3; SE FinV, 2.8; SE setback viscosity, 2.9. Means by columns with different letters show significant difference, contrast test (*p* < 0.05). BD, bulk density; EI, expansion index; WSI, water solubility index; WAI, water absorption index; MaxV, maximum viscosity; MinV, minimum viscosity; FinV, final viscosity.

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
