# Peer review of "Effects of the Addition of Flaxseed and Amaranth on the Physicochemical and Functional Properties of Instant-Extruded Products"

_foods, 2019, doi:10.3390/foods8060183_

Round 1
Reviewer 1 Report
The work by Tobias-Espinoza and co-authors is of great interest and relevance for research in the food sector. The manuscript is well written, original and contains new results that significantly advances research in the field of functional instant-extruded cereals. Introduction displays a large knowledge of the literature and the aims of the work are clearly stated. The research methodology is correct and the experimental plan carefully set. Results are clear and well discussed. The paper is appropriate for “Foods”, but English revisions are recommended.
Please, find some minor suggestions for improvement:
Abstract:
In general, the English is poor.
l. 27: description of mixtures can be improved.
Materials and Methods:
l. 92-93: “All the starting material were analysed….. “: for what? The authors may want to complete the sentence, or delete it, since more details are given in paragraph 2.4.1. Proximate analysis.
l. 130: replace “analysis” with “analyses”.
Paragraph 2.3.1. (l. 103-115). Please, improve the description of mixtures formulations. The statement in l. 105-106 is rather unclear, confusing and disjointed by the descriptors shown in Table 1.
Author Response
Dear reviewer,
We really appreciate all your comments which all of them are valuable and very helpful to improve our manuscript, we have studied them carefully and have considered your suggestions. The changes are in red in the corrected manuscript version.
In addition to all the comments and suggestions made, some extra changes were made, which are also marked in red inside the manuscript and tables. Also, a revision of the English was performed.
The following responses summarize the main revision in the manuscript.
Reviewer #1:
Abstract:
In general, the English is poor.
l. 27: description of mixtures can be improved.
Authors:
Thank you for your suggestions and comments, they were considered in the manuscript and they were red marked. The English was improved through all the manuscript. In the abstract section the description of mixtures and results were improved for a better understanding.
Reviewer #1:
Materials and Methods:
l. 92-93: “All the starting material were analysed….. “: for what? The authors may want to complete the sentence, or delete it, since more details are given in paragraph 2.4.1. Proximate analysis.
Authors:
Your observation was attended, and the sentence suggested was deleted.
Reviewer #1:
l. 130: replace “analysis” with “analyses”.
Authors:
The misspelling was corrected (L.127).
Reviewer #1:
Paragraph 2.3.1. (l. 103-115). Please, improve the description of mixtures formulations. The statement in l. 105-106 is rather unclear, confusing and disjointed by the descriptors shown in Table 1.
Authors:
Thanks, the paragraph 2.3.1 was rewritten for clarification the description of mixtures formulation (L.108-113). Also, Table 1 was modified for better understanding of the proportions of the ingredients in the mixtures.
Reviewer 2 Report
The paper addresses an important topic: factors affecting the processing of healthier (high fiber, especially soluble fiber) products with attractive texture/ properties for consumers by adding substantial amounts of high fiber materials to maize grits. However, important information is missing and should be added and other info should be modified.
1. It is essential to provided data for processing and product properties of a 100% maize grit product, without any addition. Only then a good comparison can be made of (dis-)advantages of the described products.
2. Data should be provided for levels of resistant starch - both before and after processing. (Note: the Megazyme company is providing suitable test kits). This is a highly relevant theme in (extrusion) processing of high-starch products. I am happy to note that this is already mentioned in the present version
3. The current definition of fiber (Codex, 2009) includes in addition to the high molecular weight fibers measured with the old AOAC methods 985.29 and 991.43 also low MW soluble fibres. It is likely that the levels of these low MW fibres will increase due to extrusion. Therefore it would be much better when total fibre and its fractions insoluble fibre, high MW soluble fiber and low MW soluble fiber will be measured according to AOAC Method 2011.25
Too much emphasis is given to the level of soluble fibers and its health benefits. Approved health claims in the USA and Europe related to maintaining low cholesterol levels show that minimum levels per portion of food should be 1 g for beta-glucans and much higher levels for other high MW fibers. A portion of breakfast cereals for adults is internationally agreed to be between 30 and 45g. Even the highest soluble fiber level obtained - 1.9% - provides less than 1% fiber for 45g of product.
A more suitable health related factor that can be communicated is when the addition of flax or amaranth provides at least 30% more total fiber than the basic product with maize grits only.
Author Response
Dear reviewer,
We really appreciate all your comments which all of them are valuable and very helpful to improve our manuscript, we have studied them carefully and have considered your suggestions. The changes are in red in the corrected manuscript version.
In addition to all the comments and suggestions made, some extra changes were made, which are also marked in red inside the manuscript and tables. Also, a revision of the English was performed.
The following responses summarize the main revision in the manuscript.
Reviewer #2:
1. It is essential to provided data for processing and product properties of a 100% maize grit product, without any addition. Only then a good comparison can be made of (dis-)advantages of the described products.
Authors:
Your observation is important. The 100% maize grits extruded was not used as a mixture control since the objective of this study was to evaluate the effect of the addition of ingredients that provide a high content of dietary fiber, mainly soluble fiber in the physicochemical properties on the extruded product. Therefore, the maize grits proportion was used as an ingredient to fix the maximum amount of fat (5.6%) and fiber (2.5) in all mixtures.
Reviewer #2:
2. Data should be provided for levels of resistant starch - both before and after processing. (Note: the Megazyme company is providing suitable test kits). This is a highly relevant theme in (extrusion) processing of high-starch products. I am happy to note that this is already mentioned in the present version
Authors:
Thanks for your suggestion, unfortunately the resistant starch was not directly evaluated; however, in an implicit way is considered in the dietary fiber content evaluated in the extrudate products.
Reviewer #2:
3. The current definition of fiber (Codex, 2009) includes in addition to the high molecular weight fibers measured with the old AOAC methods 985.29 and 991.43 also low MW soluble fibres. It is likely that the levels of these low MW fibres will increase due to extrusion. Therefore it would be much better when total fibre and its fractions insoluble fibre, high MW soluble fiber and low MW soluble fiber will be measured according to AOAC Method 2011.25.
Authors:
Thanks a lot for your recommendation to improve our research, this suggestion would be considered in future research.
Reviewer #2:
Too much emphasis is given to the level of soluble fibers and its health benefits. Approved health claims in the USA and Europe related to maintaining low cholesterol levels show that minimum levels per portion of food should be 1 g for beta-glucans and much higher levels for other high MW fibers. A portion of breakfast cereals for adults is internationally agreed to be between 30 and 45g. Even the highest soluble fiber level obtained - 1.9% - provides less than 1% fiber for 45g of product.
A more suitable health related factor that can be communicated is when the addition of flax or amaranth provides at least 30% more total fiber than the basic product with maize grits only.
Authors:
We appreciate your comments about the actual recommended dietary allowance (RDA) for soluble fiber, which is very important for product development as breakfast cereals and to achieve the impact in health benefits of consumers, however in this study the addition of ingredients such as amaranth and flaxseed was to evaluate the soluble fiber content and their impact in physicochemical properties of the extruded products. Although, the extruded products obtained do not reach the RDA of soluble fiber, they can contribute to the recommended daily value.
Reviewer 3 Report
The manuscript presents effects of the addition of flaxseed and amaranth on the physicochemical and functional properties of instant-extruded cereals. The study is presented in a lucid manner, yet one point need to be further explained :
Did Authors optimize the paramethers of extrussion proces (screw speed, temperature etc)?
Author Response
Dear reviewer,
We really appreciate all your comments which all of them are valuable and very helpful to improve our manuscript, we have studied them carefully and have considered your suggestions. The changes are in red in the corrected manuscript version.
In addition to all the comments and suggestions made, some extra changes were made, which are also marked in red inside the manuscript and tables. Also, a revision of the English was performed.
The following responses summarize the main revision in the manuscript.
Reviewer #3:
Did Authors optimize the paramethers of extrussion proces (screw speed, temperature etc)?
Authors:
Thanks for your comments. In this study, the extrusion conditions were not optimized. The conditions were set up based on preliminary experiments, considering as criteria; continuous flow of the extruder with some suitable expansion in the extruded products. These criteria allow us to set up: screw speed, barrel and die temperature, and feed moisture content. Conditions that are described in the section 2.3.2 Extrusion process (L.114-124).
Reviewer 4 Report
P3/L10 What percentage of minor ingredients (sucralose, cocoa and cinnamon) were added to corn grits base formula?
P4/L120 Extrusion process parameters such as moisture content of melt, screw speed, feed rate, barrel temperature profile etc. Describe the condition and the reason why selected these condition.
P6/L210 “Is important to highlight the protein content of amaranth” check it?
L317 Expansion index and bulk density is generally corelated negatively, but I can not find the the trend between two parameters.
L242 Dietary fiber content of raw materials and extrudates need to compare in one table or figure. But explain why total fiber contents decreased after extrusion process. Increase in soluble fiber after extrusion is reasonable.
L390 Explain the importance of maximum viscosity at 95C of paste viscosity of extrudate powder, there are no highest peak of paste viscosity profiles. Initial cold viscosity, minimum, and final viscosity were affected by difference formula.
Table 3. Mean ± standard error is not shown on mean.
Author Response
Dear reviewer,
We really appreciate all your comments which all of them are valuable and very helpful to improve our manuscript, we have studied them carefully and have considered your suggestions. The changes are in red in the corrected manuscript version.
In addition to all the comments and suggestions made, some extra changes were made, which are also marked in red inside the manuscript and tables. Also, a revision of the English was performed.
The following responses summarize the main revision in the manuscript.
Reviewer #4:
P3/L10 What percentage of minor ingredients (sucralose, cocoa and cinnamon) were added to corn grits base formula?
Authors:
Thanks for your comment. The minor ingredient represents 4.7% in the mixtures formulated. Table 1 shows the proportions of each one of the minor ingredients added (P.3).
Reviewer #4:
P4/L120 Extrusion process parameters such as moisture content of melt, screw speed, feed rate, barrel temperature profile etc. Describe the condition and the reason why selected these condition.
Authors:
Thanks for your comments. In this study, the extrusion conditions were set up based on preliminary experiments, considering as criteria; continuous flow of the extruder with some suitable expansion in the extruded products. These criteria allow us to set up: screw speed, barrel and die temperature, and feed moisture content. Also, for a better understanding the 2.3.2. Extrusion process section was modified, and specific condition data were included (L.114-124).
Reviewer #4:
P6/L210 “Is important to highlight the protein content of amaranth” check it?
Authors:
Thanks for your observation. The paragraph was rewritten for a better understanding (P6/L. 207-209).
Reviewer #4:
L317 Expansion index and bulk density is generally corelated negatively, but I can not find the the trend between two parameters.
Authors:
Thanks for your comments, effectively the expansion index (EI) and bulk density (BD) did not present a correlation trend between them. They are affected by the mixture composition, specifically in this study small variation in maize grits proportions were used (section 2.3.1 mixtures preparation); which avoid notorious changes between both parameters (EI and BD). A paragraph was rewritten for better understanding (P.10/L.334-342).
Reviewer #4:
L242 Dietary fiber content of raw materials and extrudates need to compare in one table or figure. But explain why total fiber contents decreased after extrusion process. Increase in soluble fiber after extrusion is reasonable.
Authors:
Thanks for your comments. The presentation of dietary fiber content of raw materials and extruded products data were made for easier discussion and better visual result presentation. The graphical presentation of results between extrudates samples allowed contrast the different fiber content between the extruded mixtures. An explanation about the total fiber content decrease was inserted in P.8/L.254-261.
Reviewer #4:
L390 Explain the importance of maximum viscosity at 95 ºC of paste viscosity of extrudate powder, there are no highest peak of paste viscosity profiles. Initial cold viscosity, minimum, and final viscosity were affected by difference formula.
Authors:
Thank you for your comments and observation. A paragraph was inserted in P.12/L.382-385 to explain the importance of maximum viscosity at 95 ºC of paste viscosity of the extrudates. The other viscosity parameters reported in table 4 presented small variations due mainly to the maize grits proportions in the different mixtures.
Reviewer #4:
Table 3. Mean ± standard error is not shown on mean.
Authors:
Thanks a lot. Your suggestion was made in the table 3.